# *ct2vl*: A Robust Public Resource for Converting SARS-CoV-2 Ct Values to Viral Loads

**DOI:** 10.3390/v16071057

**Published:** 2024-06-30

**Authors:** Elliot D. Hill, Fazilet Yilmaz, Cody Callahan, Alex Morgan, Annie Cheng, Jasper Braun, Ramy Arnaout

**Affiliations:** 1Beth Israel Deaconess Medical Center, Division of Clinical Pathology, Department of Pathology, Boston, MA 02215, USA; elliot.d.hill@duke.edu (E.D.H.);; 2Department of Pathology, Rhode Island Hospital, Warren Alpert Medical School of Brown University, Providence, MA 02903, USA; 3Beth Israel Deaconess Medical Center, Division of Clinical Informatics, Department of Medicine, Boston, MA 02215, USA

**Keywords:** COVID-19, Ct values, viral loads, Python, RT-qPCR, bioinformatics

## Abstract

The amount of SARS-CoV-2 in a sample is often measured using Ct values. However, the same Ct value may correspond to different viral loads on different platforms and assays, making them difficult to compare from study to study. To address this problem, we developed *ct2vl*, a Python package that converts Ct values to viral loads for any RT-qPCR assay/platform. The method is novel in that it is based on determining the maximum PCR replication efficiency, as opposed to fitting a sigmoid (S-shaped) curve relating signal to cycle number. We calibrated *ct2vl* on two FDA-approved platforms and validated its performance using reference-standard material, including sensitivity analysis. We found that *ct2vl*-predicted viral loads were highly accurate across five orders of magnitude, with 1.6-fold median error (for comparison, viral loads in clinical samples vary over 10 orders of magnitude). The package has 100% test coverage. We describe installation and usage both from the Unix command-line and from interactive Python environments. *ct2vl* is freely available via the Python Package Index (PyPI). It facilitates conversion of Ct values to viral loads for clinical investigators, basic researchers, and test developers for any RT-qPCR platform. It thus facilitates comparison among the many quantitative studies of SARS-CoV-2 by helping render observations in a natural, universal unit of measure.

## 1. Introduction

The real-time reverse-transcription polymerase chain reaction, commonly known as quantitative RT-PCR or RT-qPCR, is a standard method for testing human samples for the presence of viruses. Such viruses include HIV-1 (human immunodeficiency virus type 1), HCV (hepatitis C virus), and, since 2020, SARS-CoV-2 [1,2]. In RT-qPCR, the tiny amount of genetic material originally present in a positive patient sample is copied by a polymerase enzyme over repeat cycles. This results in exponential amplification that eventually leads to detectable amounts of genetic material [3]. The cycle number at which the detection threshold is reached is called the Ct value. Because the reaction is monitored continuously, the threshold may be crossed between cycles, leading to the alternative term fractional cycle number (FCN) [4]. The more starting material, the fewer cycles are needed for a signal to cross the threshold. Thus, the smaller the Ct value, the greater the amount of starting material.

RT-qPCR results can be reported either qualitatively (positive/negative) or quantitatively. For many other infections, quantitative results are usually reported not as a Ct value but as a viral load: the number of copies of viral genomic material present per unit volume of sample (i.e., a concentration). The most common unit is copies/mL. An important advantage of viral loads over Ct values is that viral loads are consistent across platforms. In contrast, Ct values are inconsistent, due to platform-specific differences in polymerase, amplification conditions, the signal-detection method, whether “dark cycles” are run, and other factors. (Dark cycles are cycles that are run but not counted toward the total.) Consequently, the same Ct value may correspond to different viral loads on different platforms and assays [5,6,7,8,9]. Such differences make it difficult to compare Ct values from study to study. For example, for SARS-CoV-2, a Ct value of 26 corresponds to a viral load of 100 copies/mL of viral transport media on one FDA-approved platform and nearly 500,000 copies/mL on another (Abbott Alinity m vs. m2000) [10]. This platform-to-platform variability makes Ct values difficult to interpret and can lead to mistaken conclusions about a patient’s clinical course (Figure 1). An additional advantage is the direct correlation between disease burden and viral load, as opposed to the inverse correlation with Ct.

PCR generally exhibits three phases. The first is a lag phase set by the stochasticity of polymerase first encountering template molecules (and in practice, the platform’s detection threshold; exponential increase occurs before the detection threshold but is not visible). The second is an exponential (or “log”) phase during which the amount of product roughly doubles each cycle. The third and final phase is a plateau due to inhibition of the enzyme by the (now-copious) product the reaction has produced. A detection threshold is crossed during the exponential phase; in fact, at least one large diagnostics company determines the threshold, and thereby the Ct or FCN, from the cycle at which maximum exponential growth is observed [4]. In the absence of additional considerations, converting from Ct to viral load would involve simply fitting this relationship using an S-shaped function (e.g., Gompertz’ growth curve). 

In practice, fitting S-shaped curves is not straightforward. A good fit requires careful weighting of datapoints in different parts of the curve. In practice, no S-shaped function—Gompertz, sigmoid, logistic, or Chapman—precisely captures the part of the curve that is the most important for viral load determination, the exponential phase [11]. Moreover, there is no guarantee that the details of the PCR formulation (e.g., multiple targets; internal controls) will not affect the details of the fit. Therefore, an alternative approach is to fit only the key determinant of the region of interest: the decrease in maximum polymerase efficiency that is frequently observed with increasing cycle number. This is especially useful in situations where template is admixed with an internal-control template. This is because internal-control template can compete with the target template, inhibiting polymerase in more complex ways than a model of a single template might capture.

In the scramble to create, approve, and validate tests for COVID-19, most SARS-CoV-2 RT-qPCR tests were not validated to output viral load [12]. Yet viral loads can be valuable indicators of where a patient is in the course of infection, as well as the likelihood of being infectious [13,14,15]. For these reasons, it has been our experience that clinical staff often informally ask laboratorians to report Ct values for their patients outside of the health record. The difficulty in interpreting Ct values has persisted into the third year after the start of the pandemic. As time goes on, the likelihood that a patient or caregiver will encounter RT-qPCR results from platforms with disparate Ct-value scales can only increase, increasing the chance of diagnostic error. 

Fortunately, the correspondence between Ct value and viral load in RT-qPCR is well understood mathematically. The validation studies that laboratories must perform in order to bring a test online can provide the data necessary to convert from Ct to viral load on clinical samples [16]. In past work, we wrote computer code to convert from Ct values to viral loads to help reveal and quantify the range of viral burden in the patient population [17]. This also allowed comparison of the sensitivity and utility of testing from different anatomical sources quantitatively and in a generalizable fashion (e.g., saliva vs. nasal secretions vs. nasopharyngeal secretions) [10,18]. We later expanded on this code to provide viral loads for new platforms brought online at our institution. However, this code applied only to our own platforms, despite SARS-CoV-2 viral loads being a global need. To address this need, here we present a much-expanded new Python package called *ct2vl* intended to make it straightforward to convert from Ct values to viral loads on any platform.

## 2. Materials and Methods

### 2.1. Ethics Statement

This work was approved as exempt by the Institutional Review Board of the Beth Israel Deaconess Medical Center (protocol 2023P000845). As exempt research, no consent was required.

### 2.2. Mathematical Derivation

Both the equation we used and its experimental validation have been described in detail in previous work [17], so we review the approach only briefly here. 

Exponential growth with decreasing replication rate yields the following equation for viral load, *v*_0_, as a function of Ct value (see the supplementary information of [17] for the derivation):(1)$$\rho \left(\beta \right)=X\beta$$
$$\log v_{0}=\log v_{L}+\int_{0}^{C_{t_{L}}}\log\rho \left(\beta \right)dx-\int_{0}^{C_{t}}\log\rho \left(\beta \right)dx$$

Here, $v_{L}$ and $C_{t_{L}}$ give the anchor point: the simplest anchor point is to let $v_{L}$ be the limit of detection (LoD) [17] and $C_{t_{L}}$ be the Ct value at the LoD. *ρ* is a polynomial fit of maximum replication rate vs. cycle at maximum replication rate (a slight change from [17]). These constitute the parameters of this model (see Implementation, below). Maximum replication rate and the cycle at maximum replication rate are derived from time series data of the form amount of material vs. cycle number (Figure 2). The amount of material is most often measured as a fluorescence intensity (e.g., of an intercalating fluorophore present in the RT-qPCR reaction mix).

### 2.3. Implementation

To parameterize Equation (1), we must find the coefficient β of the polynomial regression fit between the maximum replication rate and the cycle at maximum replication rate. To calculate the maximum replication rate and cycle at maximum replication rate, a set of signal-vs.-cycle PCR time series—“traces”—for positive samples is obtained from the platform and processed as follows. The first three cycles are discarded because the initial values of PCR traces are often noisy and may interfere with the estimation of maximum replication rate. Negative signal-intensity values are considered noise and therefore set to 0. The data are smoothed, ensuring monotonic increase (PCR product cannot decrease; slight/transient decreases sometimes observed during the lag phase are attributed to signal-detector noise). These steps result in denoised traces. Examples of processed traces are plotted in Figure 3a. Even after denoising, some noisy measurements of replication rate can be observed in the early cycles (Figure 3b).

Maximum replication rate is then calculated as the largest ratio of the signal at a given cycle to the signal at the previous cycle. A polynomial regression is fit to the relationship between maximum replication rate and cycle at maximum replication rate, yielding β. The degree of the polynomial is chosen via a cross-validation grid search over degrees 1, 2, and 3 (linear, quadratic, and cubic, respectively), providing a more robust update to the description in [17]. With β estimated and $v_{L}$ and $C_{t_{L}}$ provided by the user, the integrals in Equation (1) are solved numerically by *ct2vl*. The equations are now calibrated and are ready to convert Ct values to viral loads. The user is spared interaction with the mathematics (see Usage, below).

### 2.4. Calibration and Validation Datasets

The FDA-approved Abbott Alinity m RealTime PCR assay SARS-CoV-2 RT-qPCR testing platform was used for this analysis. Results for the Abbott m2000 have been previously described [17]. To validate *ct2vl*’s accuracy, we compared the viral load predictions from *ct2vl* to a validation dataset composed of 40 Ct values and their corresponding viral loads from two independent calibration series of SARS-CoV-2 RT-qPCR viral loads. These were from the same Abbott Alinity m machines. First, we calibrated *ct2vl* on 96 positive PCR traces from one of the Alinity machines, using Equation (1) and the known (experimentally confirmed) LoD = 100 copies/mL and mean $C_{t_{L}}=37.83$ for this machine (see Usage, below). We then used *ct2vl* to convert the 48 Ct values from the validation dataset to viral loads and compared these predicted viral loads to the ground-truth viral loads in the validation dataset to estimate the prediction error. The calibration and validation datasets are provided as part of the *ct2vl* GitHub repository.

For the validation dataset, the genome copy number was based on the reference standard produced by SeraCare (AccuPlex SARS-CoV-2 Reference Material Kit, catalog number 0505-0126). This control material consists of replication-incompetent, enveloped, positive-sense, single-stranded RNA Sindbis virus into which SARS-CoV-2 PCR targets detected by Abbott SARS-CoV-2 RT-qPCR assays have been cloned. This control material was quantified by the manufacturer using digital droplet PCR and diluted into viral transport medium for analysis. 

### 2.5. Sensitivity Analysis

To estimate *ct2vl*’s sensitivity to the $C_{t_{L}}$ parameter, we replaced the mean value of 37.83 with each of 23 different $C_{t_{L}}$ measurements from separate calibration tests. (For the final value, these were averaged together to obtain the mean.) This was performed while holding all other parameters fixed. We then measured the prediction error (|predicted viral load–known viral load|) on the validation dataset (for which viral load was known). We estimated *ct2vl*’s sensitivity to β as follows. First, we bootstrap-resampled our calibration data 1000 times (randomly sampling the same number of traces, with replacement). Next, we refit the polynomial regression (in Equation (2)) on each bootstrapped sample. Finally, we calculated the *ct2vl* prediction error on the validation dataset for each bootstrap sample. To estimate total confidence intervals, i.e., the cumulative effect of variation in $C_{t_{L}}$ and β, we bootstrap-refit β for each of the 23 $C_{t_{L}}$ values.

### 2.6. Code Coverage

In computer science, code coverage (or test coverage) is a measure of how much of the code is covered by test suites. Complete coverage means every line of code has been tested for proper function. For *ct2vl*, code test coverage was determined using the *pytest-cov* package.

## 3. Results

### 3.1. Python Package Overview

The Python package *ct2vl* takes RT-qPCR-reaction time series as input, to parameterize an equation describing the relationship between Ct value and viral load (Equation (1)). After this calibration step, it converts new Ct values to viral loads for a given $C_{t_{L}}$ and $v_{L}$. The package can be used as a command line tool or imported into Python programs. The package has 100% test code coverage and has been tested on macOS, Ubuntu, and Windows.

### 3.2. Calibration and Validation Results

For our calibration dataset, the coefficients of the polynomial fit between max replication rate and cycle at max replication rate were 2.27, −2.48 × 10^−2^, and 4.06 × 10^−4^ (Figure 3). For the validation dataset, predicted values demonstrated excellent agreement with observed values (Pearson’s *r* = 0.99, *p* < 0.001). The mean absolute error between predicted and observed viral load was 0.21 ± 0.28 log10 units (mean ± 2 s.d.), meaning that predicted viral loads were accurate to within 2.5 ± 1.5 fold, highly accurate considering that viral loads range over 10 orders of magnitude in SARS-CoV-2 infection [17]. Consistent with this finding, *R*^2^ was 0.97 for a linear fit between predicted and observed viral loads, with slope ~1:1, demonstrating the accuracy as well as the precision of *ct2vl* over the full range of the six orders of magnitude of available validation data (Figure 4).

### 3.3. Sensitivity Analysis Results

Regarding sensitivity to the $C_{t_{L}}$ parameter, we found an absolute prediction error of 0.25 ± 0.33 log10 units (mean ± 2 s.d.). Bootstrapping β parameters gave a mean absolute error of 0.24 ± 0.30 log10 units. Lastly, to measure overall sensitivity, varying $C_{t_{L}}$ and bootstrapping β parameters simultaneously gave a mean absolute error of 0.25 ± 0.35 log10 units (Figure 5).

### 3.4. Installation

*ct2vl* requires Python 3.7 or higher to be installed. Assuming *pip* is installed, to install *ct2vl*, at the command line, run $ pip install ct2vl

### 3.5. Command-Line Usage

There are two key functions: *calibrate* and *convert*. To calibrate *ct2vl* run
$ python3 -m ct2vl calibrate <traces> <LoD> <Ct_at_LoD>

Here, <traces> is a csv file containing the positive traces, where each row is a PCR reaction trace and each column is a time step in that trace. For example, for the example file positive_traces.csv (available on the GitHub repository; see Availability Statement below), the command would be the following:$ python3 -m ct2vl calibrate traces.csv 100.0 37.96

Once *ct2vl* has been calibrated using the above command, Ct values can be converted to viral loads by typing
$ python3 -m ct2vl convert <Ct>

One or multiple Ct values can be passed. For example, to convert a Ct value of 23.1 to a viral load with a LoD of 100 copies/mL and a corresponding Ct value of 38.73:$ python3-m ct2vl convert 23.1

The output is printed to the screen in a text table. Each line includes the row number, the LoD and Ct-at-LoD ($C_{t_{L}}$) used, the Ct value that was input, the viral load in units of copies/mL, and the viral load in log10 units:LoD Ct_at_LoD Ct viral_load log10_viral_load
 1 100.00 37.83 25.32 299427.73 5.48

To convert multiple Ct values to viral loads:$ python3 -m ct2vl convert 25.32 30.11 35.95

Output:LoD Ct_at_LoD Ct viral_load log10_viral_load1 100.00 37.83 25.32 299427.73 5.482 100.00 37.83 30.11 14030.16 4.153 100.00 37.83 35.95 336.52 2.53

Output can be saved to a file by providing a file path to the optional flag ‘--output’, like so:$ python3 -m ct2vl convert 100.0 37.83 23.1 --output viral_loads.tsv

Here, the tabular output will have been saved to a tab-delimited text file called viral_loads.tsv.

### 3.6. Python Package Usage

For users who are familiar with the Python programming language and environments, *ct2vl* can also be used programmatically as follows (note the capitalization):>>> from ct2vl.conversion import Converter>>> converter = Converter(traces=“traces.csv”, LoD=100.0, Ct_at_LoD=37.83)>>> viral_loads = converter.ct_to_viral_load(Ct=[25.3, 30.1, 35.9])>>> viral_loads[299427.732571, 14030.156190, 336.522756]

## 4. Discussion

The COVID-19 pandemic necessitated development and approval of RT-qPCR tests at a rate that outstripped the ability to widely validate conversion of results from Ct values to viral loads. As COVID-19 has become endemic, the probability that patients will receive results on multiple platforms may rise. So may the need to manage cases where viral load will need to be monitored over time—for example, in the context of lingering infection or long COVID [19,20]. *ct2vl* facilitates calculation of viral loads for any platform based on a laboratory’s own validation data. To our knowledge, no package exists with this functionality. The mathematics of RT-qPCR are more complicated for real-world clinical tests, which often contain internal controls and multiple targets, than in stripped-down experimental systems. In addition, accurate assessment of maximum efficiency is more important for viral load estimation than fitting the entirely of the curve (lag, log, and stationary phase). For these reasons, *ct2vl* concentrates on the most important cause of deviation from pure exponential growth—the fall of replication efficiency with cycle—and fits it empirically/phenomenologically. This is as opposed to shoe-horning a particular S-shaped curve (sigmoid function) from the many such curves that exist [21].

Traditionally, conversion from Ct values to viral loads has required first creating a standard curve spanning a range of viral loads at least as large as what is observed in clinical practice. However, creating standard curves can be time consuming and expensive, especially when viral loads range over as many orders of magnitude as they do with SARS-CoV-2 (≥1 billion-fold between the lowest and highest viral loads encountered in clinical practice) [17]. Fortunately, reliable Ct-to-viral load conversion can also be performed mathematically based on the well-understood biochemical principles of PCR [4,16,22]. This mathematical approach requires only (1) time series of signal vs. cycle number for positive samples and (2) an anchor point—the Ct value for a given viral load—such as what labs routinely measure, in replicate, when validating the limit of detection (LoD) before bringing a platform online.

Previous work adopting such an “assumption-free” [23] approach has been incorporated into a comprehensive set of tools for PCR analysis, available online and as a Python installation [24]. This work also identified the importance of measuring PCR efficiency from traces and the advantages over standard curves. The major difference is that *ct2vl* allows efficiency to vary over the course of the reaction, following observations that efficiency falls as the reaction proceeds [17] (which is consistent with theory).

Comparison against a calibration curve of well-described SARS-CoV-2 standards (Seracare) demonstrated excellent performance of *ct2vl*, including robustness to sensitivity analysis. Total error was less than half a log10 unit, an acceptable performance relative to the 10 log10 units over which viral loads vary in SARS-CoV-2 infection and comparable to the error in HIV viral load testing. *ct2vl* is made available free of charge, with a completely open-source codebase and 100% code test coverage, to facilitate customization and incorporation into laboratory workflows. We note that *ct2vl* is applicable to any situation in which calibration is available and polymerase replication rate does not rise with cycle number (i.e., all conventional PCR). It is hoped that it will prove useful beyond SARS-CoV-2, even as the world continues to have to manage successive waves of COVID-19 [25]. We anticipate researchers using *ct2vl* to facilitate conversion of Ct values to viral loads, especially to facilitate large-scale research studies [26].

One limitation of *ct2vl* is the requirement that users have access to traces (see Implementation). While such traces will be produced by most if not all platforms, accessing them may require interaction with specially trained technologists and possibly even the manufacturers. In our experience, clinical laboratorians will know the necessary individuals; those outside the clinical laboratories may need to navigate their organization or liaisons to manufacturers’ technical staff members to access the required data. A second limitation is that *ct2vl* is (at least as of this writing) for research but not clinical use. It is hoped that the public availability of the codebase, combined with 100% code coverage, will facilitate its adaptation for clinical purposes by healthcare workers, researchers, and/or manufacturers, subject to the appropriate regulatory oversight and approval.

## Figures and Tables

**Figure 1 viruses-16-01057-f001:**
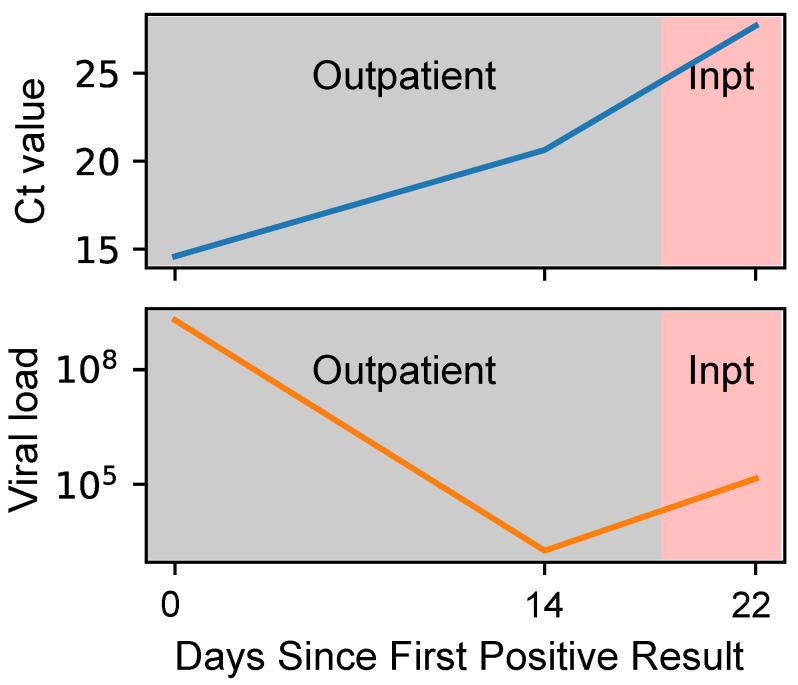
**Ct value vs. viral load (in copies/mL).** Data from a patient at our medical center. In this patient, Ct values trended consistently upward (**top panel**). Clinically, it would be reasonable to interpret this trend as reflecting a continuous fall in viral load and thereby expect clinical improvement, all other things being equal. Yet the patient worsened between Day 14 and 22, necessitating hospitalization. It happened that the Day 14 result was from a different platform to the Day 0 and Day 22 results. Consequently, the Ct value for Day 14 cannot be interpreted on the same scale as the other two datapoints; the plot is misleading. Conversion to viral load shows the true picture: a rebound in viral load, consistent with the worsening clinical picture (**bottom panel**).

**Figure 2 viruses-16-01057-f002:**
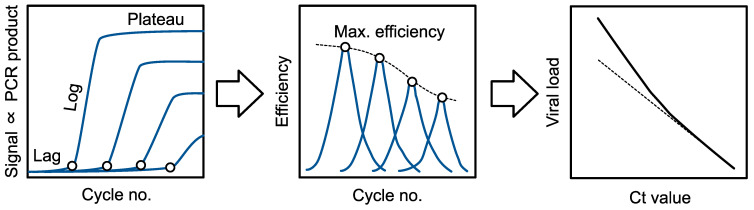
**Schematic overview of ct2vl.** Left: signal vs. cycle number traces are analyzed to find maximum replication efficiency (open circles). Middle: the fall in maximum efficiency vs. cycle number is fit by a curve (dotted line). Right: together with an anchor point (Ct at the LoD), this curve is used to convert Ct values to viral loads (solid line). Without this curve, viral load would be underestimated (dotted line).

**Figure 3 viruses-16-01057-f003:**
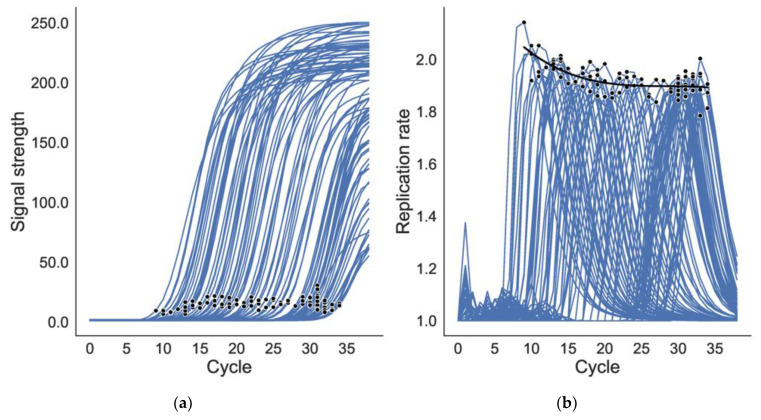
**Amplification signal and polymerase replication rate vs. RT-qPCR cycle.** (**a**) PCR traces and the points at the maximum replication rate (black dots). (**b**) The replication rate for each trace with the points where maximum replication rate is reached. Notice that the max replication rate falls as cycle increases.

**Figure 4 viruses-16-01057-f004:**
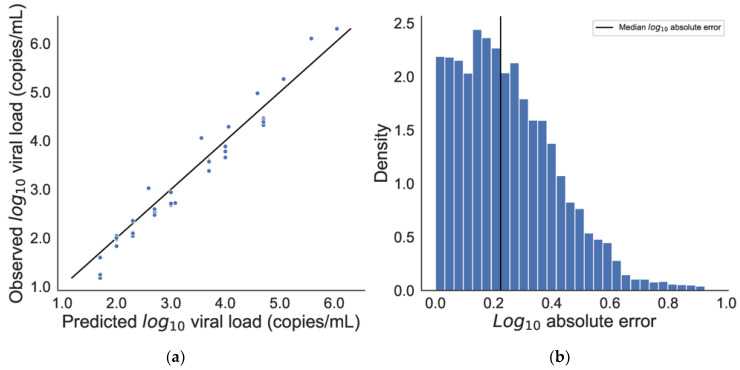
**Predicted vs. observed viral load.** (**a**) Viral loads predicted by *ct2vl* compared with observed viral loads in a validation dataset of known viral loads (Seracare). *R*^2^ = 0.96. (**b**) Histogram of absolute prediction error. Notice that the majority of the error is below half a log10 unit. Since viral loads can range over 10 orders of magnitude, this error is within a clinically useful range.

**Figure 5 viruses-16-01057-f005:**
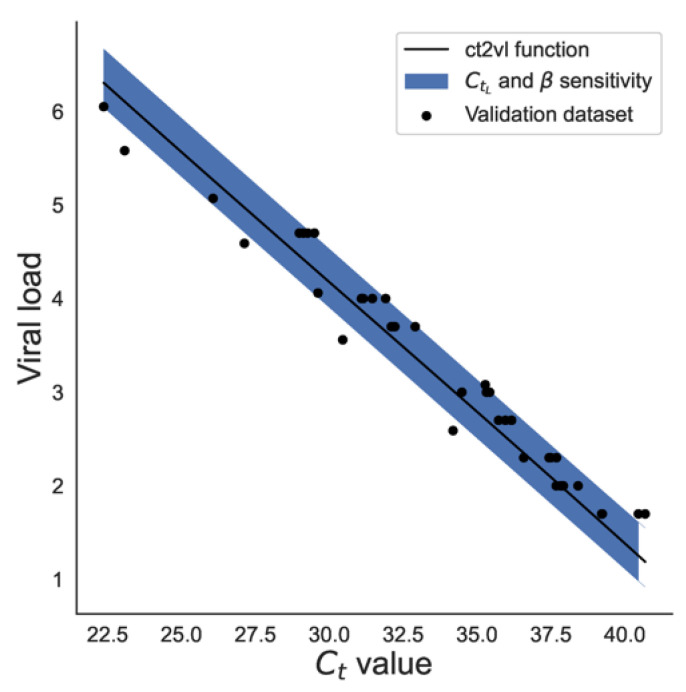
**Ct value vs. viral load.** The validation dataset (black dots) with the *ct2vl* prediction function (mean ± 2 std.) when $C_{t_{L}}$ was varied and the calibration parameters β were bootstrapped.

## Data Availability

*ct2vl* is freely available on the Python Package Index at https://pypi.org/project/ct2vl/, accessed on 26 June 2024 or via GitHub at https://github.com/ArnaoutLab/ct2vl, accessed on 26 June 2024.

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
