# Peer review of "ct2vl: A Robust Public Resource for Converting SARS-CoV-2 Ct Values to Viral Loads"

_viruses, 2024, doi:10.3390/v16071057_

Round 1

Reviewer 1 Report

Comments and Suggestions for Authors

It is valuable to publish copies instead of cq values .

This is what RDML-Analyze of the rdml-tools.com does if you select absolute quantification and the "rule of thumb" algorithm. This is not a novel approach and was presented on the qPCR conference in Freising in 2023. You can use this tool as a python library or a web interface.

If you want to calculate copies, you need the PCR efficiency. This you can calculate from raw data. I am convinced that the sigmoidal approach is not suitable, as the plateau and the transition depend on the amplicon parameters. See the work of JM Ruijter with LinRegPCR for details.

In your manuscript you have to discriminate between cq value which is a single number like 23.7 and amplification curves which have a fluorescent measurement per cycle. Cq values do not allow to calculate PCR efficiency and fitting of sigmoidal curves. So the software either does not convert cq values as claimed but evaluates raw data or is unable to do sigmoidal fitting and correction of the PCR efficiency. Please clarify.

A small thing: Also in the first phase you have exponential amplification of your amplicon, you just do not se the minute amount as it is hidden in background fluorescence.

Author Response

This is what RDML-Analyze of the rdml-tools.com does if you select absolute quantification and the "rule of thumb" algorithm. This is not a novel approach and was presented on the qPCR conference in Freising in 2023. You can use this tool as a python library or a web interface.

We thank the reviewer for the introduction to RDML, which we now cite in our manuscript (Untergasser 2021 [PMID 34433408]). We also cite Ramakers 2003 (PMID 12618301) by some of the same authors, which had useful details.

Assuming RDML-Analyze uses the approach in Ramakers 2003 (PMID 12618301), the main difference is that our method captures the changes in efficiency over the course of the reaction, whereas the prior work appears to treat efficiency as a constant. We have now clarified this in the text with a new paragraph in the Discussion (ca. lines 309-314).

In your manuscript you have to discriminate between cq value which is a single number like 23.7 and amplification curves which have a fluorescent measurement per cycle. Cq values do not allow to calculate PCR efficiency and fitting of sigmoidal curves. So the software either does not convert cq values as claimed but evaluates raw data or is unable to do sigmoidal fitting and correction of the PCR efficiency. Please clarify.

We believe the discussion of the comparison to Ramakers 2003, described above, also clarifies this point (ca. lines 311-316).

A small thing: Also in the first phase you have exponential amplification of your amplicon, you just do not se the minute amount as it is hidden in background fluorescence.

We have clarified by adding "exponential increase occurs before the detection threshold but is not visible" ca. line 62.

Reviewer 2 Report

Comments and Suggestions for Authors

During the ongoing SARS-CoV-2 pandemic, RT-qPCR tests have become highly used for both diagnosis and research purposes. However, there are multiple methods, machines, and analyses of RT-qPCR for SARS-CoV-2, and most of these tests are incapable of validating the actual viral load. To address this issue, Hill et al. developed a Python package called ct2vl. This package can convert Ct values to viral loads for any RT-qPCR platform, making it an invaluable tool for laboratories to validate virus infection and compare data from different resources. Although the package's usage is currently limited to research purposes and not clinical use, it remains a powerful tool for the research community.

Minor Comment:

1.     Page 4: There should be (Figure 2) but not (Figure 2a) in line 132.

Author Response

We thank the reviewer for the positive review of our work.

1.     Page 4: There should be (Figure 2) but not (Figure 2a) in line 132.

We have corrected this error.

Reviewer 3 Report

Comments and Suggestions for Authors

The paper by Hill et al., is a very well-written and described evaluation of a python-based code to estimate viral load for SARS-Co-V2. The evaluation metrics and approach are appropriate and provided in detail. One area of improvement could be inclusion of data comparing the author's code with standard approach e.g., delta CR for viral load evaluation in a clinical cohort. This would provide additional and practical proof-of-concept and confirm utility.

Round 2

Reviewer 1 Report

Comments and Suggestions for Authors

I have difficulties to understandwhen you user Cq values and when raw fluorescence traces. As I said before: "Cq values do not allow to calculate PCR efficiency and fitting of sigmoidal curves. So the software either does not convert cq values as claimed but evaluates raw data or is unable to do sigmoidal fitting and correction of the PCR efficiency." This essential point was not clarified at all, my review suggestions were not answered. Adding two sentences is not sufficient.

Minor points:

What is the scale of viral load in Fig. 5? virus/ml? If its the 10^x then the efficiency is really low.

The article mentions supplemental data, but I do not see any.